# Treatment Effect of CT-Guided Periradicular Injections in Context of Different Contrast Agent Distribution Patterns

**DOI:** 10.3390/diagnostics12040787

**Published:** 2022-03-23

**Authors:** Vera Reuschel, Cordula Scherlach, Christian Pfeifle, Matthias Krause, Manuel Florian Struck, Karl-Titus Hoffmann, Stefan Schob

**Affiliations:** 1Institut für Neuroradiologie, Universitätsklinikum Leipzig AöR, Liebigstr. 20, 04103 Leipzig, Germany; vera.reuschel@med.uni-goettingen.de (V.R.); cordula.scherlach@medizin.uni-leipzig.de (C.S.); karl-titus.hoffmann@medizin.uni-leipzig.de (K.-T.H.); 2Institut für Diagnostische und Interventionelle Neuroradiologie, Universitätsmedizin Göttingen, Robert-Koch-Str. 40, 37075 Göttingen, Germany; 3Klinik und Poliklinik für Orthopädie, Unfallchirurgie und Plastische Chirurgie, Universitätsklinikum Leipzig AöR, Liebigstr. 20, 04103 Leipzig, Germany; christian.pfeifle@medizin.uni-leipzig.de; 4Klinik und Poliklinik für Neurochirurgie, Universitätsklinikum Leipzig AöR, Liebigstr. 20, 04103 Leipzig, Germany; m.krause@medizin.uni-leipzig.de; 5Klinik und Poliklinik für Anästhesiologie und Intensivtherapie, Universitätsklinikum Leipzig AöR, Liebigstr. 20, 04103 Leipzig, Germany; manuelstruck@web.de; 6Abteilung für Neuroradiologie, Universitätsklinik und Poliklinik für Radiologie, Universitätsklinikum Halle (Saale) Ernst-Grube-Str. 40, 06120 Halle (Saale), Germany

**Keywords:** CT-guided percutaneous injections, spinal pain management, interventional nerve root block, contrast media distribution

## Abstract

Acutely manifesting radicular pain syndromes associated with degenerations of the lower spine are frequent ailments with a high rate of recurrence. Part of the conservative management are periradicular infiltrations of analgesics and steroids. The purpose of this study is to evaluate the dependence of the clinical efficacy of CT-guided periradicular injections on the pattern of contrast distribution and to identify the best distribution pattern that is associated with the most effective pain relief. Using a prospective study design, 161 patients were included in this study, ensuring ethical standards. Statistical analysis was performed, with the level of statistical significance set at *p* = 0.05. A total of 37.9% of patients experienced significant but not long-lasting (four weeks on average) complete pain relief. A total of 44.1% of patients experienced prolonged, subjectively satisfying pain relief of more than four weeks to three months. A total of 18% of patients had complete and sustained relief for more than six months. A significant correlation exists between circumferential, large area contrast distribution including the zone of action between the disc and affected nerve root contrast distribution pattern with excellent pain relief. Our results support the value of CT-guided contrast injection for achieving a good efficacy, and, if necessary, indicative repositioning of the needle to ensure a circumferential distribution pattern of corticosteroids for the sufficient treatment of radicular pain in degenerative spine disease.

## 1. Introduction

Low back pain (LBP) with a radicular component in the vast majority of cases represents a clinical manifestation of degenerative spine disease. Despite the fact that most patients presenting with radicular back pain belong to this category, more serious underlying conditions, such as malignancy, infection, fractures, cardiovascular disease and cauda equina syndrome must be ruled out before standard treatment is initiated [1,2]. Differentiating between the degenerative spine continuum and the above-mentioned serious spinal pathologies, however, is not trivial. For this purpose, rather specific combinations of red flags for each of those conditions have been determined with the aim to help distinguishing critical from non-critical conditions. In the case of cauda equina syndrome, for example, important red flags are stool and urinary incontinence, saddle anesthesia and loss of sphincter tone [2]. The identification of such red flags must trigger further diagnostics, most importantly, sufficient imaging, to verify the diagnosis and determine the need for emergency surgery. After excluding those serious spinal pathologies in the individual case, the common therapeutic algorithm for radicular back pain syndromes can be pursued. However, the natural history of disabling, acutely manifested lumbar and lumbosacral disc disease lacking distinct neurological deficits is not uniform, and clear indications for operative intervention, minimally invasive therapy or purely conservative management cannot be established from the literature [3].

In general, approximately 70% of patients suffering from radiating LBP without further neurological deficits caused by intervertebral disc herniation recover without surgery within six weeks [4]. However, certain symptoms are linked to unfavorable outcomes. For example, the presence of severe radicular pain has been associated with worse outcomes than the presence of back pain without a dominating radicular component, and weakness of knee extension also indicates an unfavorable further course [5].

Reviewing the findings of numerous studies in this context, for example, the large ten-year observational study of Weber and colleagues [6] and the cohort study of Kim and colleagues [7], it is important to underline that no significant differences exist when comparing outcomes of surgery with conservative management in the long term. Nevertheless, early outcomes after 1 month appear to be better after surgery [7], and early surgery may facilitate fast recovery and return to work, although post-surgical sensory-motor outcomes after one year do not differ from the outcomes of conservative therapy [8].

As a consequence, patients presenting with severe or progressive neurological deficits require surgery without question [9], whilst a conservative approach may be preferable in many cases [9].

Based on a number of magnetic resonance imaging (MRI) studies, it has become apparent that the majority of radicular conditions associated with herniated discs regress spontaneously, which explains the significant percentage of patients experiencing spontaneous relief after few weeks without treatment [10,11,12,13]. Surprisingly, especially large-volume hernias are resorbed comparatively fast [9] and show good outcomes if progressive nerve injury or cauda equina syndrome are lacking [12]. This phenomenon has been linked to the fact that the large amount of immune-privileged disc-tissue, which suddenly invades the highly immune-potent intrathecal compartment, initiates a profound inflammatory response that triggers fast neovascularization, enzymatic degradation and phagocytosis of the herniated tissue [13,14,15]. In addition, retraction of not sequestered disc tissue, together with dehydration and shrinkage of herniated nucleus, have been considered responsible for the spontaneous resorption of disc tissue.

In context of those studies, it has been postulated that patients suffering from disabling pain without major neuromuscular deficit, whose condition did not improve over a period of six weeks under conservative management, are in need of complementary treatment [16,17,18]. Aside from that, no clear treatment strategies for patients who had already undergone spine surgery for disc herniation and experienced a disabling relapse later on are available. For both groups of patients, off-label application of periradicular glucocorticoid injections has proven beneficial, especially considering pain relief and regaining functional independence as the most important early therapeutic goals [19,20]. Nevertheless, not all patients experience an equal extent of pain reduction after periradicular corticoid injections. This, on the one hand, is certainly related to the timing of the injection, as chronified nerve root lesions naturally respond less well to glucocorticoid exposure than only very recently affected nerve roots do. On the other hand, procedural protocols and the technical approach to periradicular glucocorticoid injections vary significantly between physicians. For example, a great number of percutaneous injections are performed without imaging guidance, although precise needle positioning is paramount for achieving a sufficient distribution of the anti-inflammatory agent [21]. Therefore, the use of fluoroscopy, ultrasound and computed tomography (CT) in percutaneous spine treatments has increased markedly, as they facilitate drug delivery and decrease complication rates [21]. Among those imaging modalities, CT has emerged as the preferred guiding technique for percutaneous spinal interventions, as image quality is reliable and user independent, allowing highly precise navigation of the needle in all three spatial planes, providing sufficient bony and soft tissue contrast in the region of interest and accurately predicting the spatial distribution of the pharmaceutical cocktail around the nerve root when performing a preceding contrast medium injection [21]. Despite all efforts, everyday routine reveals that sufficient post-interventional pain relief does not occur in every patient, even though most treated individuals report a satisfactory effect. In our institutional experience, the pattern of contrast agent distribution is highly predictive for treatment success.

This study, therefore, investigated the hypothesis that good clinical efficacy of CT-guided periradicular injections is not exclusively, but causally, linked to the individual pattern of contrast agent dispersion at hand and that the latter must be used to optimize the needle positioning when the perineural opacification appears insufficient. More specifically, characteristic distribution patterns were reviewed in the context of the individually experienced pain relief with the aim to identify the distribution pattern associated with most effective pain relief.

## 2. Materials and Methods

### 2.1. Ethics Statement

Our study, investigating prospectively included cases from January 2018 to December 2019, was approved by the institutional ethics committee (local IRB 075/18-ek). Informed consent of each patient regarding the scientific use of radiological and clinical data was obtained in writing, either from the patient or his/her legal representative.

### 2.2. Patients and Procedure

A total of 161 patients with lumbosacral CT-guided periradicular injections were enrolled in this study. The visual analog scale (VAS) was used as outcome measure. Sample size estimation was based on a pre-study observation of the treatment effect and eventually performed using the freely available tool G*Power, setting the significance level to 0.05 and power to 0.8, as reported earlier [22]. The patients were evaluated and treated in either the orthopedic or the neurosurgical department, where a complete diagnostic workup was performed. This included a thorough physical examination, with special attention being paid to the presence of red flags in order to rule out serious spinal conditions, as mentioned earlier [1,2]. Additionally, plain radiography and MRI of the lumbar spine were performed prior to the intervention. Patients who were evaluated to potentially benefit from periradicular infiltrations were then referred to the department for neuroradiology. Inclusion criteria were acute or chronic LBP with a characteristic radicular component corresponding to the affected nerve root dermatome. Exclusion criteria were apparent myelopathy, acute fractures or hemorrhage and spinal infections.

In all cases, the periradicular infiltration was performed under CT-guidance using a Philips Brilliance Big Bore, equipped with 16 detector rows. The patient was placed in prone position for the intervention. An unenhanced scan of the spinal level of interest was performed using a priorly mounted radiopaque skin marker for further reference. The trajectory and depth required for the intra-foraminal navigation of the needle were evaluated using a few cross-sectional CT images at the level of interest. The paravertebral puncture site was marked on the skin surface. After surgical disinfection, the procedure was performed under sterile conditions.

Using CT control images for close to real time navigation, a 22 G (0.7 mm) spinal needle was safely placed in the posterior part of the neuro-foramen. Negative needle aspiration—to avoid intravascular injections—was followed by the injection of approximately 1 mL of contrast agent (Solutrast 250 M, Bracco Imaging, Berlin, Germany).

### 2.3. Assessment of Technical Success

The contrast agent distribution pattern in each intervention was assessed by two experienced radiologists (VR, SS) in a consensus approach, according to the following categories:without primarily apparent nerve root contact or;spatially confined, non-circumferential spread along the nerve root or;extensive surface contact with at least partially circumferential distribution, including the intraspinal portion of the nerve root.

In case of discrepancies (n = 13), the distribution pattern was re-evaluated together. All of those 13 cases had discrepant assessments as follows: one reader evaluated spatially confined, non-circumferential spread, whereas the other reader evaluated at least partially circumferential distribution. Consent was reached in all cases, and the distribution pattern was eventually rated as spatially confined, non-circumferential.

Figure 1 (left) shows a representative case of a 57-year-old male patient suffering from acute onset predominantly radicular LBP of the left-handed side, corresponding to the L5 root dermatome (VAS prior injection: 8) for two weeks. The final needle position was achieved per primam without the need for further corrections. Extensive continuous periradicular and epidural opacification (level 3) was achieved. The patient reported an immediate pain relief (VAS 0), which endured over the six-month period.

Figure 1 (middle) shows a representative case of a 49-year-old male patient suffering from acute onset predominantly radicular LBP of the right-handed side, corresponding to the L5 root dermatome (VAS prior injection: 7) for eight weeks. Needle placement was performed without difficulties. Spatially confined contrast agent dispersion along the nerve root (level 2) was achieved. The patient reported a moderate pain relief (VAS 4) after approximately 8 h, which endured for six weeks only.

Figure 1 (right) shows a representative case of a 58-year-old male patient suffering from acute onset predominantly radicular LBP of the left-handed side, corresponding to the L5 root dermatome (VAS prior injection: 7) for three weeks. Despite unchallenging anatomical circumstances, needle placement was difficult due to anxiety and agitation of the patient. Peripheral contrast agent dispersion without an immediately apparent nerve root contact was achieved. The patient reported no significant pain relief (VAS 6) after 24 h.

After the desired needle position was achieved and the contrast agent distribution was rated to be sufficient (or in few cases could not be further improved due to limiting factors, such as anatomic barriers, needle length or patient’s discomfort and pain), a combination of 1 mL Triamcinolonacetonid 40 mg (Volon A 40, Dermapharm AG, Grünwald Germany) and 2 mL Bupivacainhydrochlorid 0.5% (Carbostesin, AstraZeneca, Wedel, Germany) was applied. Subsequently, each patient was instructed to maintain bed rest until the next day, mainly to avoid falls related to bupivacaine-induced transient paresis.

Treatment efficacy was assessed using a visual analog scale (VAS)-based questionnaire at two time points; first, 24 h after injection, and at routinely performed follow-up appointments between 3 months and up to 12 months post-procedure. All patients had a long-term follow up after CT-guided percutaneous infiltration as follows: 15.5% by telephone interview and 84.5% via in-hospital appointment for reassessment.

### 2.4. Statistical Analysis

The strategy of the statistical analysis was reviewed and approved by Dr. Gussew, which we thankfully acknowledge. The analysis was performed using SPSS 23.0, with the level of statistical significance set at *p* = 0.05.

#### 2.4.1. Whole Collective Analysis

First of all, Gaussian vs. non-Gaussian distribution of the data was assessed. Related to the comparatively heterogeneous composition of our patient collective, group comparisons (performing Kruskal–Wallis test) and correlative analysis (Spearman’s Rho calculation) were performed using the whole dataset in the first step. Additionally, the following subgroups were formed and analyzed further.

#### 2.4.2. Subgrouping according to First-Line Therapy

Patients who received pharmaceutical therapy only, pharmaceutical and physical therapy or surgery as initial treatment were analyzed as a separate cohort. For each subgroup, the treatment effect according to VAS and technical success were assessed independently.

Furthermore, correlative analysis to elucidate the relation between treatment success according to VAS and technical–procedural success was performed.

#### 2.4.3. Subgrouping according to the Duration of Predominantly Radicular Backpain Prior to CT-Guided Percutaneous Therapy

Patients who had acute (<6 weeks), subacute (6 weeks–6 months) and chronic pain (>6 months) prior to CT-guided percutaneous infiltration were each analyzed as separate subgroups. For each subgroup, the treatment effect according to VAS and technical success were assessed independently.

## 3. Results

### 3.1. General Findings and Remarks

The individually evaluated, distinct pain syndromes of the included patients are summarized in the following categories:29.8% had LBP accompanied by a diffuse radiating component;55.3% had LBP and dermatome-related radiating pain with matching dysesthesia;14.3% had LBP and dermatome-related radiating pain with additional motor deficits. In greater detail, of those 23 patients:8.07% (n = 13) had a L4 syndrome with accompanying knee extensor muscle weakness;6.21% (n = 10) had a L5 syndrome with accompanying foot drop;0.6% (one patient) suffered from dermatome-matching dysesthesia only.

In three cases of known allergy to iodinated contrast agent, we refrained from using contrast agent. The clinical and demographic characteristics of 161 patients are summarized in Table 1. All patients received spinal MRI prior to CT-guided percutaneous infiltration. MRI findings and correspondingly treated spinal segments are summarized in Table 2 and Table 3. Table 4 summarizes the technical success, Table 5 shows the clinical effect of the treatment.

A total of 18.6% of patients underwent micro-discectomy after a conservative approach was chosen initially but did not result in sufficient pain relief. For reasons of clarity, only the first CT-guided percutaneous infiltration in each of the 161 patients was evaluated, although a number of individuals were treated repeatedly. On average, 1.5 injections/patient were performed (range 1–5 injections/patient). The treatment success of the CT-guided percutaneous infiltrations is presented for the overall collective and subgroups under the respective headings, as follows.

### 3.2. Overall Group Comparison and Correlative Analysis

As pre- and post-interventional VAS scores, as well as contrast agent distribution pattern, emerged to be not normally distributed, Kruskal–Wallis was used for group comparisons and Spearman’s Rho was performed for correlative analysis. Kruskal–Wallis test revealed a statistically significant difference (*p* = 0.001) in clinical success reflected by VAS decrease when comparing the three contrast agent distribution patterns as grouping variables. Figure 2a graphically summarizes the difference, employing the conventional boxplots. Furthermore, Spearman’s Rho showed a moderately strong association (r = 0.555, *p* = 0.001) between technical success (contrast agent distribution pattern) and treatment efficacy (post-treatment decrease in VAS score). Figure 2b visualizes the association between contrast agent distribution and the effect on VAS.

Additionally, the duration of maintained pain relief was evaluated with the following results: 37.9% of patients experienced a significant but not long-lasting (on average four weeks) complete pain relief. A total of 44.1% of patients experienced a longer, subjectively satisfying pain reduction of more than four weeks, up to three months. A total of 18% of patients had complete and persisting relief of more than six months.

### 3.3. Subgrouping according to Temporal Extent of Predominantly Radicular Backpain Prior to CT-Guided Percutaneous Therapy

#### 3.3.1. Group Comparison and Correlative Analysis in Subgroups Accounting for Different First-Line Therapies

Kruskal–Wallis was used for group comparisons, and Spearman’s Rho was performed for correlative analysis. Kruskal–Wallis test revealed a statistically significant difference (*p* = 0.001) in clinical success reflected by VAS decrease when comparing the three distinct first-line therapies as a distinguishing variable. Figure 3 graphically summarizes the difference in treatment efficacy between the three first-line therapy groups employing conventional boxplots. Correspondingly, Spearman’s Rho showed comparatively strong associations between technical success (contrast agent distribution) and therapeutic effect (VAS decrease) in the pharmaceutical-only group (r = 0.587, *p* = 0.001), pharmaceutical + physical therapy group (r = 0.564, *p* < 0.001) but only a relatively weak association (r = 0.384, *p* = 0.006) in the surgery-first group.

#### 3.3.2. Subgroup Comparison and Correlative Analysis in Subgroups accounting for Pain Duration between Onset and CT-Guided Percutaneous Treatment

Kruskal–Wallis was used for group comparisons, and Spearman’s Rho was performed for correlative analysis. Kruskal–Wallis test revealed a statistically significant difference (*p* < 0.001) in clinical success reflected by VAS decrease when comparing the three distinct first-line therapies as a distinguishing variable. Figure 4 graphically summarizes the difference in treatment efficacy between the three acuity-groups employing conventional boxplots. Correspondingly, Spearman’s Rho showed comparatively strong associations between technical success (contrast agent distribution) and therapeutic effect (VAS decrease) in the acute group (r = 0.511, *p* < 0.001), subacute group (r = 0.622, *p* < 0.001) but only a relatively weak association (r = 0.536, *p* < 0.001) in the chronic group.

## 4. Discussion

This study demonstrates a significant correlation between contrast agent distribution pattern, indicating local drug distribution, and pain relief after periradicular infiltration.

To the best of our knowledge, evidence regarding the association between different distribution patterns of contrast agent in CT-guided percutaneous infiltrations of lumbar or sacral nerve roots, which are compromised by herniated discs or osteoligamentous degeneration, is scarce, and prospective investigations, in particular, are lacking.

Furthermore, the distinct subset of patients who had prior spinal surgery for severely disabling disc herniation and presented again with a similar syndrome has not yet been included in systematic investigations exploring the efficacy of percutaneous CT-guided infiltrations. In this context, our study provides important insight on the value of minimally invasive CT-guided procedures for pain treatment in previously operated patients.

Although class I evidence for the superiority of image-guided percutaneous infiltrations over conservative medical treatment for pain management remains wanted, a number of meaningful retrospective studies have demonstrated the great value of the minimally invasive treatment for patients suffering from radiating back pain due to lumbosacral neural impingement [23,24,25]. As a consequence, periradicular injections have found broad acceptance in daily practice and are performed with increasing frequency [26]. However, regarding the procedural details of image-guided spinal nerve root treatments, no consensus but a variety of very different protocols exist [27,28]. For example, epidurography is seldom used in the office practice setting [29], as its superior accuracy claims greater procedural duration, whereas the highly cost-efficient “loss of resistance” technique does not require contrast injections but definitely results in a distinctly greater number of inaccurate needle-tip positions [30]. Thus, although requiring greater procedural duration, contrast medium injections performed in order to establish and verify an ideal needle position that guarantees the best possible distribution of the anti-inflammatory pharmaceutical, are nowadays recommended and demanded for good outcomes [31].

Interestingly, a number of previous studies failed to show a significant correlation between the quality of contrast agent distribution and clinical effect in image-guided percutaneous infiltrations [32,33]. However, it is comprehensible that a precise delivery of the pharmaceutical agent to the site of neural impingement is pivotal for treatment efficacy and most certainly facilitated by high-resolution three-dimensional imaging of the anatomical target [34,35,36].

In accordance with this hypothesis, our results demonstrate a direct, significant correlation between treatment efficacy and quality of contrast agent distribution in CT-guided percutaneous infiltrations. More specifically, our study indicates that circumferential, extensive contrast agent distribution, including the impact zone between the herniated disc and the affected nerve root, as such representing the best achievable level of contrast agent distribution, is associated with excellent pain relief and is, per se, largely independent of long pre-existing pain duration or even episodes of prior surgery. This corresponds well with the main findings of the few previously published investigations, which found an extensive, perineural contrast agent dispersion in fluoroscopy to be associated with good clinical results when performing transforaminal corticoid injections [34,35].

The local inflammatory response, initiated at the very moment the immune-privileged disc content extrudes into the immunologically unrestricted epidural space, is an important component of the resorption process of the herniated disc tissue [35]. In detail, monocyte–macrophage recruitment for phagocytic degradation, together with macrophage-triggered neovascularization, are the main players of the first-line inflammatory reaction that synergistically contributes to the cleansing of herniated nucleus pulposus fragments [15,16]. Despite its importance for the resorption of herniated disc tissue, the inflammatory response can have detrimental effects when involving the affected nerve root [35]. Both mechanisms—direct mechanical pressure exerted by the herniated disc, as well as secondary macrophage-triggered chemotaxis and neovascularization—induce blood–nerve barrier breakdown and perineural edema, which in turn induce a self-sustaining and self-enhancing local inflammatory cascade that culminates in a local autoimmune-response with infiltration of peripheral immune cells, not only into the affected nerve root, but also the adjacent spinal cord followed by persisting activation of spinal-resident microglia [36,37,38,39]. The sum of those pathophysiological events per se causes acute pain and—if not counterbalanced sufficiently—is responsible for increasingly irreversible chronification of the latter [35]. Therefore, early and precise application of corticosteroids, which quickly restore the disturbed blood–nerve–brain barrier and thus limit the otherwise vicious circle of intra-epineural leukotaxis, neo-angiogenesis and inflammatory remodeling, is pivotal for treatment success [40,41,42].

In light of the latter investigations, the results of our study validate and substantiate the necessity of contrast agent injection, and if necessary, repositioning of the needle to achieve a continuous circumferential distribution pattern, prior to periradicular application of corticosteroids for treatment of LBP in herniated disc disease. In general, a precise and early treatment is recommendable. On the one hand, it achieves early pain relief and enables the patient to perform synergistic physical therapy; on the other hand, it restores blood–nerve barrier integrity at the earliest possible time point, which prevents inflammatory nerve root remodeling, and thus, chronified radicular pain.

Our results emphasize that, although symptoms may be chronified in patients suffering from osteoligamentous degeneration, they still benefit from periradicular infiltrations. As mentioned above, a precise periradicular drug delivery represses local inflammation, reduces perineural edema and initiates pain relief, also in chronic pain, due to permanent nerve compression [43,44]. Previous studies confirm these findings and emphasize that mobility and physical activity improve due to pain relief in symptomatic chronic degeneration [45]. Additionally, access to physiotherapy may prevent upcoming spinal surgery [46,47].

Somewhat unexpectedly, periradicular infiltrations in patients who had preceding spinal surgery also yield respectable results and should not be considered a strictly palliative approach only. Previously damaged nerve roots may already have undergone inflammatory remodeling to some extent; however, suffering a second disc herniation induces recurrence of the inflammatory cascade and most certainly requires local anti-inflammatory treatment to contain and at least partially reverse the structural inflammatory conversion of the affected spinal segment. Additionally, as suggested quite recently by Moen et al., patients who suffered lumbar disc herniation exhibit a distinctly pro-inflammatory serum profile even one year after herniation, which strongly suggests a persisting local and systemic inflammatory condition in the aftermath of the neural impingement [44].

Recent investigations indicate that periradicular infiltrations using ozone as therapeutic agent also bear great potential [48,49]. Ozone has quite a similar set of analgesic and anti-inflammatory effects compared to corticosteroids but lacks the unfavorable side effects of the latter [49]. Although ozone may be slightly less effective according to some investigators [48], further studies should compare both agents, especially in combination with the contrast distribution assessment proposed in our work.


*Take-home points:*
application of contrast agent in periradicular infiltrations is helpful for improving the distribution of corticosteroids along the affected nerve root;a circumferential distribution is associated with good treatment efficacy;periradicular infiltrations, as a complementary treatment for both conservative therapy and surgery, help to significantly reduce pain for the majority of patients in the early phase and potentially up to six months, only a minority experiences permanent and sufficient pain reliefin comparison to the natural course, periradicular infiltrations improve outcomes and are a safe, effective and, therefore, justifiable complementary treatment.


*Limitations:* Our study, although performed with a prospective design, is limited by the clinical heterogeneity of the included patients, the small number of included patients and the small final number of patients in each subgroup. More specifically, the assessment of the overall extent of spinal degeneration is not reflected well in our study. A superior way of appropriately addressing this aspect in a future study is the scoring proposed by Eksi et al. [50], who not only considered the disc and the osseous elements but also included spinal musculature into their concept. Furthermore, in order to maintain a comprehensible study setup and a required level of simplicity, the individual pain medications, their duration, psychosocial factors, patient satisfaction and past experiences were not evaluated and require attention in further studies.

## Figures and Tables

**Figure 1 diagnostics-12-00787-f001:**
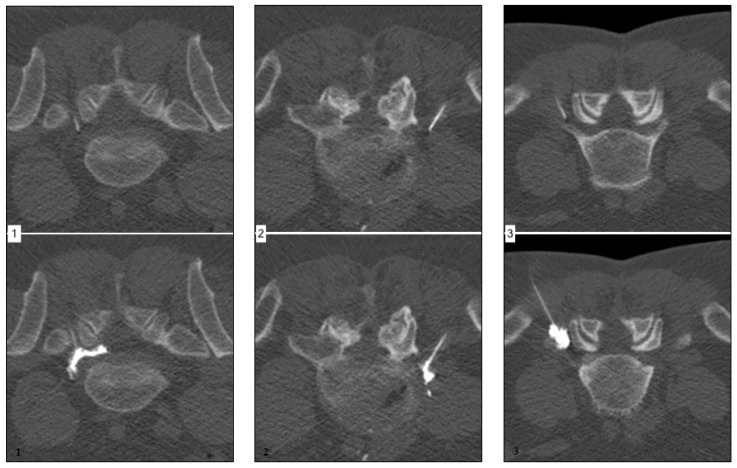
Corresponding examples of contrast agent distribution pattern.

**Figure 2 diagnostics-12-00787-f002:**
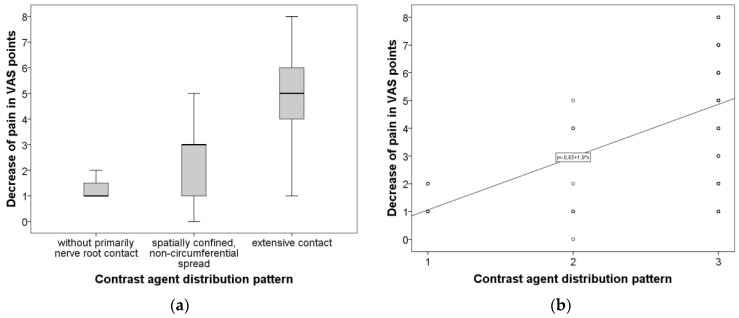
(**a**) Difference of VAS decrease comparing the three groups of contrast agent distribution pattern. Graphically summarizes the difference of VAS decrease comparing the three groups of contrast agent distribution pattern employing conventional boxplots. (**b**) Demonstrates the association between contrast agent distribution and effect on VAS. Graphically summarizes the difference of VAS decrease comparing the three groups of contrast agent distribution pattern employing conventional boxplots.

**Figure 3 diagnostics-12-00787-f003:**
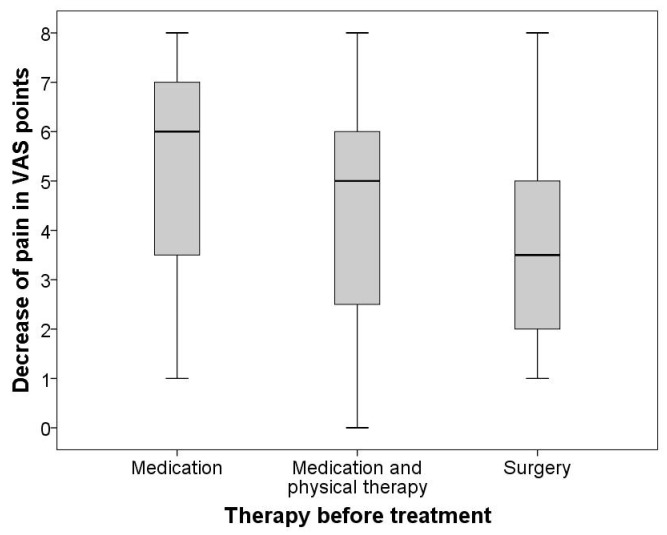
Difference in treatment efficacy between the three first-line therapy groups. Graphically summarizes the difference in treatment efficacy between the three first-line therapy groups employing conventional boxplots.

**Figure 4 diagnostics-12-00787-f004:**
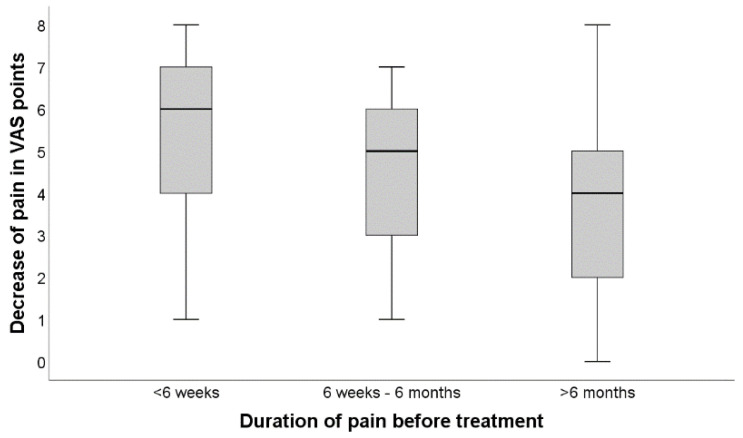
Difference in treatment efficacy between the three acuity-groups. Demonstrates the difference in treatment efficacy between the three acuity-groups employing conventional boxplots.

**Table 1 diagnostics-12-00787-t001:** Baseline characteristics in the overall collective.

Age	58.69 (15.1); Range 20–83
Gender	90 male; 71 female
BMI	66 normal; 70 overweight; 25 obese
Working ability	21.7% yes; 42.9% no; 35.4% retired
Treatment strategy prior to PRT	19.3% pain medication only 49.7% meds and physio 31.1% surgery
Current mobility	74.5% unaffected; 25.5% limited

**Table 2 diagnostics-12-00787-t002:** MRI findings in the overall collective.

	N	%
Disc herniation (bulge or protrusion)	58	36
Osteoligamentous degeneration (e.g., FH, LSS ^1^)	41	25.5
Degeneration and disc herniation	51	31.7
Spondylolisthesis	6	3.7
Multisegmental herniations	1	0.6
Degenerative aggravation in context of old fracture in osteoporosis	2	1.2
Post-traumatic (fracture) kyphosis	1	0.6
No causative MRI finding for LBP	1	0.6

^1^ FH: Facet Hypertrophy, LSS: Lumbar Spinal Stenosis.

**Table 3 diagnostics-12-00787-t003:** Spinal nerve root segment treated with CT-guided percutaneous therapy.

Segment	N	%	Left	Right
L2	6	3.7	3	3
L3	18	11.2	9	9
L4	23	14.3	10	13
L5	78	48.4	43	35
S1	36	22.4	18	18

**Table 4 diagnostics-12-00787-t004:** Technical success in the overall collective.

Contrast Distribution Pattern	N	%
Without primarily apparent nerve root contact	15	9.3
Spatially confined, non-circumferential spread along the nerve root	17	10.6
Extensive surface contact with at least partially circumferential distribution, including the intra-spinal portion of the nerve root and epidural space	126	78.3

**Table 5 diagnostics-12-00787-t005:** Summary of short-term pain relief in VAS points.

	Median	Range
VAS before treatment	6.91	2–9
VAS after treatment	2.68	0–9
Difference after PRT in VAS points	4.30	0–8

## Data Availability

All relevant data are presented in the manuscript. Further inquiries can be made and will be answered by the first author of the paper.

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
