# Peer review of "Treatment Effect of CT-Guided Periradicular Injections in Context of Different Contrast Agent Distribution Patterns"

_diagnostics, 2022, doi:10.3390/diagnostics12040787_

Round 1

Reviewer 1 Report

Authors present a prospective study on 161 patients to evaluate the 
dependence of the clinical efficacy of CT-guided periradicular injections on the pattern of contrast  distribution and to identify the best distribution pattern that is associated with the most effective pain relief. Authors report that 37.9% of patients experienced significant but not long-lasting (four weeks on average) complete pain relief, 44.1% experienced prolonged, subjectively satisfying pain relief of more  four weeks- 3 months, while 18% had complete and sustained relief for more than six months. A significant 
correlation was found between circumferential, large area contrast distribution including the zone of action between the disc and affected nerve root contrast distribution pattern with  pain relief.31Repositioning of the needle to ensure a circumferential distribution pattern of corticosteroids for the sufficient treatment of radicular pain in degenerative spine disease has been suggested. 

Several remarks:

-Introduction: there is no significant differences in outcomes when comparing surgery 
with conservative management in the long term ... post-surgical sensory-motor outcomes after one
year do not differ from the outcomes of conservative therapy - If this sentence is used, please clarify it thoroughly. This may apply only for patients with herniated discs who have only pain and no neurological deficits; it is definitely not true for patients who experience deficits, urinary or stool incontinence, cauda-conus symptoms etc. Introduction seems to be written in favor of the periradicular intervention, downsizing the role of surgical therapy. With all respect to periradicular intervention therapy, this modality of treatment is only an addition to the conservative treatment with medication and to surgical therapy. From this most important issue evolves also the misinterpretation of the results - "A total of 18.6 % underwent micro-discectomy as a result of insufficient conservative 
medical treatment including CT-guided percutaneous infiltrations". CT-guided infiltrations are often needed in case when a) patient has several different findings which could explain the symptoms, so that periradicular infiltration which leads to relief of symptoms is indicative for a problem at the certain level; b) patient does not want a surgical therapy and then the level/the nerve root which is most likely to cause symptoms gets infiltrated. For majority of patients which a spine surgeon treats, CT-guided infiltration is more diagnostic than therapeutic measure. The true question would be - how many patients with clear indication for surgery according to neurosurgeon and orthopedic surgeon rejected the proposed surgical treatment, got a periradicular infiltration instead and got better afterwards. I suggest to include this aspect in the Introduction and interpretation of the results. 

I suggest to include following references on this matter:

Azharuddin A, Aryandono T, Magetsari R, Dwiprahasto I. Predictors of the conservative management outcomes in patients with lumbar herniated nucleus pulposus: A prospective study in Indonesia. Asian J Surg. 2022 Jan;45(1):277-283. doi: 10.1016/j.asjsur.2021.05.015. Epub 2021 Aug 9. PMID: 34384675.

Ma Z, Yu P, Jiang H, Li X, Qian X, Yu Z, Zhu Y, Liu J. Conservative Treatment for Giant Lumbar Disc Herniation: Clinical Study in 409 Cases. Pain Physician. 2021 Aug;24(5):E639-E648. PMID: 34323452.

Kim CH, Choi Y, Chung CK, Kim KJ, Shin DA, Park YK, Kwon WK, Yang SH, Lee CH, Park SB, Kim ES, Hong H, Cho Y. Nonsurgical treatment outcomes for surgical candidates with lumbar disc herniation: a comprehensive cohort study. Sci Rep. 2021 Feb 16;11(1):3931. doi: 10.1038/s41598-021-83471-y. PMID: 33594185; PMCID: PMC7887235.

Figures 1-3 need to be labeled, the way they are now it is unclear if you have to look at the 6 radiological scans vertical or parallel. 

Is the circumferential distribution of the contrast in experts opinion a sign that the infiltration was performed in the aimed area? Does the contrast distribution without nerve contact or with non-circumferential spread indicated that these infiltrations were not done properly, i.e. that the nerve root was not infiltrated? Did the patients with prior surgery have better or worse outcome following infiltration compared to patients without surgery? Data on type and duration of medication are missing - this is especially important if these were patients with chronic pain syndrome or with use of opioids. 

Drawback of the study is low number of patients, please include this in limitations. 

For Discussion I suggest to include and discuss following references:

Krahulik D, Vaverka M, Hrabalek L, Pohlodek D, Jablonsky J, Valosek J, Zapletalova J. Periradicular corticosteroid infiltration for radicular pain - comparison of Diprophos and Depomedrone and ozone effects. Biomed Pap Med Fac Univ Palacky Olomouc Czech Repub. 2021 Nov 15. doi: 10.5507/bp.2021.061. Epub ahead of print. PMID: 34782796.

SucuoÄŸlu H, SoydaÅŸ N. Does paravertebral ozone injection have efficacy as an additional treatment for acute lumbar disc herniation? A randomized, double-blind, placebo-controlled study. J Back Musculoskelet Rehabil. 2021;34(5):725-733. doi: 10.3233/BMR-200194. PMID: 33843663.

Author Response

Dear reviewer 1,

we want to express our great gratitude for the opportunity to improve the quality of our manuscript „Treatment effect of CT-guided periradicular injections in context of different Contrast Agent Distribution Patterns“, which we submitted to diagnostics. We appreciate all the time and effort that you invested in our work.

Each of your comments was addressed accordingly, the manuscript was subsequently modified and a point to point response is provided as follows.

Several remarks:

-Introduction: there is no significant differences in outcomes when comparing surgery 
with conservative management in the long term ... post-surgical sensory-motor outcomes after one
year do not differ from the outcomes of conservative therapy - If this sentence is used, please clarify it thoroughly. This may apply only for patients with herniated discs who have only pain and no neurological deficits; it is definitely not true for patients who experience deficits, urinary or stool incontinence, cauda-conus symptoms etc.

We thank you for this comment. The introduction was rephrased according your suggestion.

Introduction seems to be written in favor of the periradicular intervention, downsizing the role of surgical therapy. With all respect to periradicular intervention therapy, this modality of treatment is only an addition to the conservative treatment with medication and to surgical therapy.

Dear reviewer 1, downsizing the role of surgical therapy as important modality for the management of disc disease is not intended, we thank you for this comment. We agree with you, periradicular interventional therapy must be considered an adjunct treatment and cannot substitute surgery. In accordance with your comment, the introduction was changed and the discussion amended.

From this most important issue evolves also the misinterpretation of the results - "A total of 18.6 % underwent micro-discectomy as a result of insufficient conservative 
medical treatment including CT-guided percutaneous infiltrations". CT-guided infiltrations are often needed in case when a) patient has several different findings which could explain the symptoms, so that periradicular infiltration which leads to relief of symptoms is indicative for a problem at the certain level; b) patient does not want a surgical therapy and then the level/the nerve root which is most likely to cause symptoms gets infiltrated. For majority of patients which a spine surgeon treats, CT-guided infiltration is more diagnostic than therapeutic measure. The true question would be - how many patients with clear indication for surgery according to neurosurgeon and orthopedic surgeon rejected the proposed surgical treatment, got a periradicular infiltration instead and got better afterwards. I suggest to include this aspect in the Introduction and interpretation of the results. 

Dear reviewer 1, we agree with your comment. In those respective patients, a conservative approach including periradicular injections as complementary treatment had been chosen, but did not result in a sufficient pain relief. The passage was rewritten accordingly. The intention of our manuscript is not to recommend periradicular infiltrations as a substitute, but to reveal their complementary therapeutic potential and – in our opinion – to increase their efficacy via the assessment of the contrast distribution pattern.

We furthermore agree with your `true question´, which was well addressed by the proposed reference of Kim et al. – this is a very good angle to this and will be pursued in a further study.

I suggest to include following references on this matter:

Azharuddin A, Aryandono T, Magetsari R, Dwiprahasto I. Predictors of the conservative management outcomes in patients with lumbar herniated nucleus pulposus: A prospective study in Indonesia. Asian J Surg. 2022 Jan;45(1):277-283. doi: 10.1016/j.asjsur.2021.05.015. Epub 2021 Aug 9. PMID: 34384675.

Ma Z, Yu P, Jiang H, Li X, Qian X, Yu Z, Zhu Y, Liu J. Conservative Treatment for Giant Lumbar Disc Herniation: Clinical Study in 409 Cases. Pain Physician. 2021 Aug;24(5):E639-E648. PMID: 34323452.

Kim CH, Choi Y, Chung CK, Kim KJ, Shin DA, Park YK, Kwon WK, Yang SH, Lee CH, Park SB, Kim ES, Hong H, Cho Y. Nonsurgical treatment outcomes for surgical candidates with lumbar disc herniation: a comprehensive cohort study. Sci Rep. 2021 Feb 16;11(1):3931. doi: 10.1038/s41598-021-83471-y. PMID: 33594185; PMCID: PMC7887235.

Dear reviewer 1, the references were included accordingly.

Figures 1-3 need to be labeled, the way they are now it is unclear if you have to look at the 6 radiological scans vertical or parallel. 

Figures 1-3 were labelled according to your suggestion.

Is the circumferential distribution of the contrast in experts opinion a sign that the infiltration was performed in the aimed area? Does the contrast distribution without nerve contact or with non-circumferential spread indicated that these infiltrations were not done properly, i.e. that the nerve root was not infiltrated?

Dear reviewer 1, this is exactly the case. The position of the needle within the neuroforamen may appear perfect, even for very experienced interventionists, but the contrast distribution pattern tells how the corticoid will (or will not) distribute along the nerve root. In cases of a circumferential distribution, the efficacy of the procedure is definitely superior compared to the other categories.

Did the patients with prior surgery have better or worse outcome following infiltration compared to patients without surgery?

In our study, in patients with prior surgery the efficacy of the infiltrations was the lowest compared to the other first line treatment groups. Please also see section 3.3.1 and Figure 5 in this regard.

Data on type and duration of medication are missing - this is especially important if these were patients with chronic pain syndrome or with use of opioids. 

Dear reviewer 1,we agree with you, this is a limitation of our prospective study. We initially discussed, whether to evaluate the individual pain medication of each patient or not, and decided to only use sufficiently comparable categories. Therefore, we only included the general use of pain medication in our questionaire a category. The limitations section was amended accordingly.

Drawback of the study is low number of patients, please include this in limitations. 

The limitations section was amended accordingly

For Discussion I suggest to include and discuss following references:

Krahulik D, Vaverka M, Hrabalek L, Pohlodek D, Jablonsky J, Valosek J, Zapletalova J. Periradicular corticosteroid infiltration for radicular pain - comparison of Diprophos and Depomedrone and ozone effects. Biomed Pap Med Fac Univ Palacky Olomouc Czech Repub. 2021 Nov 15. doi: 10.5507/bp.2021.061. Epub ahead of print. PMID: 34782796.

SucuoÄŸlu H, SoydaÅŸ N. Does paravertebral ozone injection have efficacy as an additional treatment for acute lumbar disc herniation? A randomized, double-blind, placebo-controlled study. J Back Musculoskelet Rehabil. 2021;34(5):725-733. doi: 10.3233/BMR-200194. PMID: 33843663.

The suggested references are now included and discussed in the revised version of the manuscript.

We again want to express our gratitude for your constructive criticism. We are looking forward to answering any further question that you may have.

Reviewer 2 Report

Dear Author,

I've read the paper entitled "Treatment effect of CT-guided periradicular injections in context of different Contrast Agent Distribution Patterns" with great interest.

The authors postulated  that the outcomes of CT-guided periradicular injections is primarily depending on the individual pattern of contrast agent dispersion, and one must try to correct the needle positioning in case of unsatisfying perineural opacification.  In detail, characteristic distribution patterns were reviewed with the aim of whether the distribution pattern associated with effective pain relief or not.  The results of the study support the value of CT-guided contrast injection for achieving a good efficacy, and, if necessary, indicative repositioning of the needle to ensure a circumferential distribution pattern of corticosteroids for the sufficient treatment of radicular pain in degenerative spine disease.

I've some concerns regarding to paper that are detailed below:

Major Concerns:
1. As the study group showed heterogeneity in clinical and radiological findings albeit this was presented as a limitation, it'd be better to cluster those parameters or classify the group by a scoring system that includes spinal degeneration patterns, ie. "Proposal for a new scoring system for spinal degeneration: Mo-Fi-Disc (https://doi.org/10.1016/j.clineuro.2020.106120)" in order not to compare apples and pears.  By this was - if possible - preoperative radiological degenerative status'd be clarified and distribution pattern of the contrast mediom on CT scan'd better be evaluated.

2. As this study primarily discusses contrast agent distribution pattern on CT scans, intra- and inter-reliability study was not performed.  I think, such evaluation should may affect the scientific level of the study, if possible (if not, please discuss this issue as a limitation).  

 Minor Concerns:

  1. p1 l39 - Better to understand "lumbar and lumbosacral disc diseases" instead of "...lumbar and sacral disc disease...".
  2. Statistically significance level of 0.05 is the most commonly accepted. However, it’s the analyst’s responsibility to determine how much evidence to require for concluding that an effect exists.  So please check the significance level in abstract and in the main text as it is presented as "...the level of statistical significance set at P < 005." 

Thank you,

Respectfully yours,

Author Response

Dear reviewer 2,

we want to thank you for the opportunity to improve our manuscript „Treatment effect of CT-guided periradicular injections in context of different Contrast Agent Distribution Patterns“, which we submitted to diagnostics for publication as an original study. We appreciate your constructive criticism and all the time and effort that you invested in our manuscript. Each of your comments was addressed accordingly, the manuscript was subsequently modified and a point to point response is provided as follows.

Major Concerns:
1. As the study group showed heterogeneity in clinical and radiological findings albeit this was presented as a limitation, it'd be better to cluster those parameters or classify the group by a scoring system that includes spinal degeneration patterns, ie. "Proposal for a new scoring system for spinal degeneration: Mo-Fi-Disc (https://doi.org/10.1016/j.clineuro.2020.106120)" in order not to compare apples and pears.  By this was - if possible - preoperative radiological degenerative status'd be clarified and distribution pattern of the contrast mediom on CT scan'd better be evaluated.

Dear reviewer 2, thank you for introducing the very interesting article and the suggestion to evaluate spinal degeneration according to the proposed classification system, which is indeed a superior approach, as it includes all the relevant elements involved in spinal degeneration. However, considering the basic idea of our study and the already significant amount of data presented in our manuscript, we would like to pursue your suggestion in a future study, that primarily uses the classification of Prof. Eksi and colleagues for subgrouping.
We amended the limitations section and discussion accordingly, and included the proposed reference.

With regards to the idea of evaluating the contrast agent distribution in a spiral scan we agree with you, it would be a better method to assess the contrast agent distribution compared to single slices. However, lumbar spine CT is associated with a significant exposure to radiation, which is why refrained from doing an extra scan for the assessment, as the additional x-ray exposure is not justifiable by the comparatively littly advantage gained through this additional scan.

  1. As this study primarily discusses contrast agent distribution pattern on CT scans, intra- and inter-reliability study was not performed.  I think, such evaluation should may affect the scientific level of the study, if possible (if not, please discuss this issue as a limitation).  

Dear reviewer 2, we absolutely agree with your comment, information in this regard was not presented in the previous version of the manuscript. The rating was performed in a consensus approach. The materials and methods section was amended accordingly.

 Minor Concerns:

  1. p1 l39 - Better to understand "lumbar and lumbosacral disc diseases" instead of "...lumbar and sacral disc disease...".

Dear reviewer 2, the passage was rewritten accordingly.

  1. Statistically significance level of 0.05 is the most commonly accepted. However, it’s the analyst’s responsibility to determine how much evidence to require for concluding that an effect exists.  So please check the significance level in abstract and in the main text as it is presented as "...the level of statistical significance set at P < 005." 

Dear reviewer 2, the significance level was set to 0.05. The abstract and the main text were corrected accoringly.

We again want to express our gratitude for your constructive criticism. We are looking forward to answering any further question that you may have.

Reviewer 3 Report

Dear 
I realize that the authors have many journals to consider when they want to publish their work, so I appreciate your interest in Diagnostics journal; I am very sorry not to be able to write in a more positive way.
It is evident that you have put a great deal of effort into this project and I want to praise your efforts,
Fortunately, the actual contribution from your study is clear and, the manuscript as currently written suggests that it might be suitable for sharing information about this interesting topic on the management  of radicular pain syndromes but the manuscript that you reported, needs any majors edits.
I should like to thank you for give me an opportunity to consider this work for publication.
It may be that the you would like to consider resubmitting it, in which case I hope that the comments from my review may help you to revise it before resubmitting it. These comments are given below.
Best Regards

- Introduction section: few references are missing in the introduction; moreover, differential diagnosis is never mentioned: some patients may have red flags for serious pathologies, to insert some sentences and references: 
# Maselli F, Palladino M, Barbari V, Storari L, Rossettini G, Testa M. The diagnostic value of Red Flags in thoracolumbar pain: a systematic review. Disabil Rehabil. 2020 Aug 19:1-17.
#Finucane LM, Downie A, Mercer C, Greenhalgh SM, Boissonnault WG, Pool-Goudzwaard AL, Beneciuk JM, Leech RL, Selfe J. International Framework for Red Flags for Potential Serious Spinal Pathologies. J Orthop Sports Phys Ther. 2020 Jul;50(7):350-372
- Materials and Methods: Patients & procedure: you should provide additional information on how the patient was preassigned to your attention; again no mention is made of differential evaluation, only neurological examination is described, to insert some sentences and references already reported
- Materials and Methods: in the statistics section, enter a few sentences about how you calculated the sample size useful for the study
- results: better describe the subgroups of patients [for example: what do you mean and what were the motor deficits found]; 
You better describe whether or not there were adverse events;
- in discussion section: Discussions should be reviewed in light of the overall improvement of the paper. Redundant sentences and prewritten information should be avoided. Focus on take-home messages and how that information impacts the clinical practice of management these patients. Moreover in limitations describe the differences found between the natural history of the disease and subjects who have improved their condition in four to six weeks, what would make the treatment you propose safe, effective and justifiable; add a few sentences of why you did not take into account any outcome measures of psychosocial factors, patient satisfaction, past experiences or their opinions.

minor edits: 
- some words are incorrect or truncated [example sciatica] or you had already used them before and then repeated with acronym [example MRI] use the acronym the first time the word appears;
- low back pain is used many times, replace with acronym LBP;

Author Response

Dear reviewer 3,

we want to thank you for the opportunity to improve our manuscript „Treatment effect of CT-guided periradicular injections in context of different Contrast Agent Distribution Patterns“, which we submitted to diagnostics for publication as an original study. We appreciate your constructive criticism and all the time and effort that you invested in our manuscript. Each of your comments was addressed accordingly, the manuscript was subsequently modified and a point to point response is provided as follows.

Reviewer 3

Dear 
I realize that the authors have many journals to consider when they want to publish their work, so I appreciate your interest in Diagnostics journal; I am very sorry not to be able to write in a more positive way.
It is evident that you have put a great deal of effort into this project and I want to praise your efforts,
Fortunately, the actual contribution from your study is clear and, the manuscript as currently written suggests that it might be suitable for sharing information about this interesting topic on the management  of radicular pain syndromes but the manuscript that you reported, needs any majors edits.
I should like to thank you for give me an opportunity to consider this work for publication.
It may be that the you would like to consider resubmitting it, in which case I hope that the comments from my review may help you to revise it before resubmitting it. These comments are given below.
Best Regards

Introduction section: few references are missing in the introduction; moreover, differential diagnosis is never mentioned: some patients may have red flags for serious pathologies, to insert some sentences and references: 

# Maselli F, Palladino M, Barbari V, Storari L, Rossettini G, Testa M. The diagnostic value of Red Flags in thoracolumbar pain: a systematic review. Disabil Rehabil. 2020 Aug 19:1-17.
#Finucane LM, Downie A, Mercer C, Greenhalgh SM, Boissonnault WG, Pool-Goudzwaard AL, Beneciuk JM, Leech RL, Selfe J. International Framework for Red Flags for Potential Serious Spinal Pathologies. J Orthop Sports Phys Ther. 2020 Jul;50(7):350-372

Dear reviewer 3, thank you for this valuable suggestion. The introduction was rewritten in accordance with your comment, both references were included.

- Materials and Methods: Patients & procedure: you should provide additional information on how the patient was preassigned to your attention; again no mention is made of differential evaluation, only neurological examination is described, to insert some sentences and references already reported

Dear reviewer 3, the materials and methods section was revised accordingly.

- Materials and Methods: in the statistics section, enter a few sentences about how you calculated the sample size useful for the study- results: better describe the subgroups of patients [for example: what do you mean and what were the motor deficits found]; 

Dear reviewer 3, the manuscript was amended according to your suggestions.

-You better describe whether or not there were adverse events;

Dear reviewer 3, there were no adverse events in our study. The results were amended accordingly.

in discussion section: Discussions should be reviewed in light of the overall improvement of the paper. Redundant sentences and prewritten information should be avoided. Focus on take-home messages and how that information impacts the clinical practice of management these patients.

Dear reviewer 3, the discussion was reviewed and shortened where appropriate. Take home messages were included at the end of the discussion.

Moreover in limitations describe the differences found between the natural history of the disease and subjects who have improved their condition in four to six weeks, what would make the treatment you propose safe, effective and justifiable; add a few sentences of why you did not take into account any outcome measures of psychosocial factors, patient satisfaction, past experiences or their opinions.

Dear reviewer 3, the limitations were amended according to your suggestions.

minor edits: 
- some words are incorrect or truncated [example sciatica] or you had already used them before and then repeated with acronym [example MRI] use the acronym the first time the word appears;
- low back pain is used many times, replace with acronym LBP;

Dear reviewer 3, the minor edits were performed as you suggested.

We again want to express our gratitude for your constructive criticism. We are looking forward to answering any further question that you may have.

Round 2

Reviewer 1 Report

The authors have sufficiently answered the reviewers remarks.

Author Response

Dear reviewer 1,

 thank you for reviewing the revised version of our manuscript.

The authors have sufficiently answered the reviewers remarks.

Dear reviewer 1. We again want to thank you for your valuable contribution to improve our manuscript.

Reviewer 2 Report

Dear Editor,
I've read the revised paper entitled "Treatment effect of CT-guided periradicular injections in context of different Contrast Agent Distribution Patterns" with great interest.

I'd like to thank the authors for their effort to leverage the scientific level of their manuscript.

As I've mentioned before, as the study group is heterogeneic, in order not to compare - apples and pears -, it'd be scientifically better for the manuscript to be evaluated by a statistician.

Thank you,

Respectfully yours,

Author Response

Dear reviewer 2,

thank you for reviewing the revised version of our manuscript.

Dear Editor,
I've read the revised paper entitled "Treatment effect of CT-guided periradicular injections in context of different Contrast Agent Distribution Patterns" with great interest.

I'd like to thank the authors for their effort to leverage the scientific level of their manuscript.

As I've mentioned before, as the study group is heterogeneic, in order not to compare - apples and pears -, it'd be scientifically better for the manuscript to be evaluated by a statistician.

Thank you,

Respectfully yours,

Dear reviewer 2. The strategy of the statistical analysis was reviewed and approved by our statistician, which is accordingly stated in our manuscript.

However, we agree with your comment – as addressed in the first revision – a more comprehensive evaluation of the overall degeneration of the lumbar spine can be performed using the Mo-Fi-Disc classification, which the must be included in the statistical analysis. As this is beyond the scope of our study, this will be addressed in a further investigation.

Reviewer 3 Report

Dear Authors

I should like to thank you for give me an opportunity to consider this work for publication. You well done the a point by point answer to the comments of the reviewers.

Minor note:

- In 2.2. Patients & procedure

A total of 161 patients with lumbosacral CT-guided periradicular injections were 130 enrolled in this study. The visual analog scale was [missing acronym VAS]

Author Response

Dear reviewer 3,

thank you for reviewing the revised version of our manuscript. The remaining comment was addressed appropriately as outlined below:

Dear Authors

I should like to thank you for give me an opportunity to consider this work for publication. You well done the a point by point answer to the comments of the reviewers.

Minor note:

- In 2.2. Patients & procedure

A total of 161 patients with lumbosacral CT-guided periradicular injections were 130 enrolled in this study. The visual analog scale was [missing acronym VAS]

The acronym is now implemented in the updated version of the manuscript.
